# Comparative Cytological and Transcriptome Analyses of Anther Development in *Nsa* Cytoplasmic Male Sterile (1258A) and Maintainer Lines in *Brassica napus* Produced by Distant Hybridization

**DOI:** 10.3390/ijms23042004

**Published:** 2022-02-11

**Authors:** Man Xing, Chunyun Guan, Mei Guan

**Affiliations:** 1Hunan Branch of National Oilseed Crops Improvement Center, Changsha 410128, China; xingman1897@gmail.com (M.X.); oilguancy@gmail.com (C.G.); 2College of Agriculture, Hunan Agricultural University, Changsha 410128, China; 3Southern Regional Collaborative Innovation Center for Grain and Oil Crops in China, Changsha 410128, China

**Keywords:** *Brassica napus*, cytoplasmic male sterility, pollen abortion, transcriptome, energy metabolism

## Abstract

1258A is a new line of *B.napus* with *Nsa* cytoplasmic male sterility (CMS) with potential applications in hybrid rapeseed breeding. Sterile cytoplasm was obtained from XinJiang *Sinapis arvensis* through distant hybridization and then backcrossed with 1258B for many generations. However, the characteristics and molecular mechanisms underlying pollen abortion in this sterile line are poorly understood. In this study, a cytological analysis revealed normal microsporogenesis and uninucleate pollen grain formation. Pollen abortion was due to non-programmed cell death in the tapetum and the inability of microspores to develop into mature pollen grains. Sucrose, soluble sugar, and adenosine triphosphate (ATP) contents during microspore development were lower than those of the maintainer line, along with an insufficient energy supply, reduced antioxidant enzyme activity, and substantial malondialdehyde (MDA) accumulation in the anthers. Transcriptome analysis revealed that genes involved in secondary metabolite biosynthesis, glutathione metabolism, phenylpropane biosynthesis, cyanoamino acid metabolism, starch and sucrose metabolism, and glycerolipid metabolism may contribute to pollen abortion. The down regulation of nine cytochrome P450 monooxygenases genes were closely associated with pollen abortion. These results suggest that pollen abortion in 1258A CMS stems from abnormalities in the chorioallantoic membranes, energy deficiencies, and dysfunctional antioxidant systems in the anthers. Our results provide insight into the molecular mechanism underlying pollen abortion in *Nsa* CMS and provide a theoretical basis for better heterosis utilization in *B.napus*.

## 1. Introduction

Cytoplasmic male sterility (CMS) prevents self-pollination through pollen abortion. It is important for the utilization of crop heterosis, which has led to substantial improvements in crop growth and development as well as resistance to adverse conditions and disease, leading to substantial increases in crop yields. Hybrid breeding is one of the most effective methods for increasing food production in various crops [1]. Currently, heterosis is widely used in rice [2], maize [3], rape [4], sunflower [5], tomato [6], and other crops, and a large number of hybrid seeds are used in production.

*Brassica napus* has strong heterosis, and three-line hybrids are widely used in production. Commonly used CMS systems for *B.napus* are *pol* CMS and *Shaan2A* CMS [7,8]. In addition to these, CMS systems used in *B.napus* production are *nap* CMS [9], *Ogu* CMS [10], *hau* CMS [11], *NCa* CMS [12], *SaNa-1A* CMS [13], and *Nsa* CMS [14]. There are many types of CMS in oilseed rape, which can be divided into three categories according to their cytoplasmic origin: (1) intraspecific hybridization or mutations, (2) distant hybridization, and (3) cell fusions. For example, the *B.napus* cytoplasmic sterile lines *pol* CMS and *nap* CMS both harbor natural mutations in the mitochondrial genome. The sterility gene *orf224*, which controls *pol* CMS, is a chimeric gene located upstream of *atp6* and is co-transcribed with *atp6* [15,16]. The sterility gene in *B.napus nap* CMS is *orf222*, which encodes a protein similar in structure to that of the *orf224* gene, another cause of CMS in *B.napus* [15]. Sterile cytoplasm of *hau* CMS in *B.napus* originates from *Brassica juncea* and is transferred to *B.napus* by intraspecific hybridization [11]. *Ogu* CMS is selected by distant interspecific hybridization; *Ogu* CMS is the first CMS type found in radish [17] and is obtained by crosses and consecutive backcrosses with *B.napus*. The application of protoplast fusion techniques for somatic cell crosses has resulted in the recombination of parental mitochondrial genomes, which may lead to the creation of new mitochondrial sterility genes. *Nsa* CMS is a novel heterologous CMS system in *B.napus* obtained from somatic hybridization of *B.napus* and *S.arvensis* [18]. *Orf346* is a mitochondrial gene causing CMS in *Nsa* CMS; it leads to a disruption in ATP production and ROS accumulation during pollen development [19].

The physiological characteristics of male sterile anthers represent an important link between male sterility at the molecular level and the individual regulation of fertility, providing an in-depth understanding of the mechanism underlying male sterility in plants. For example, many years ago, researchers discovered that an insufficient sugar supply during anther development leads to male sterility [20]. In maize cytoplasmic male sterile lines, starch biosynthesis during pollen maturity is related to changes in the expression patterns of many sugar metabolism genes [21]. This suggests that sugar biosynthesis or transporter genes are associated with pollen development in CMS plants. Normal anther development cannot be achieved without an adequate energy supply, and mitochondria are important for energy metabolism. The presence of sterile genes in *B.napus* CMS mitochondria, which are capable of producing CMS proteins leading to defective mitochondrial function, results in unmet energy requirements during male reproductive development. These sterility genes include *orf224*, *orf138*, *orf222*, *orf263*, *orf288*, and *orf346* [15,16,22,23,24,25]. The mitochondrial *orf346* gene can cause a reduction in ATP during pollen development, resulting in an insufficient energy supply as well as excessive ROS accumulation and thus pollen abortion [19]. During the normal physiological metabolic activities of plants, moderate amounts of ROS, such as H_2_O_2_, are produced; however, they are cleared in a timely manner by the organism [25]. Insufficient clearance leads to the accumulation of malondialdehyde (MDA), disrupting the physiological and biochemical functions of relevant tissues [26]. Plants contain reactive oxygen scavenging systems, including superoxide dismutase (SOD), peroxidase (POD), and catalase (CAT), to eliminate ROS damage [27,28]. Therefore, the physiological and biochemical characteristics of the anthers of sterile and maintenance lines are important for studying the mechanism underlying CMS.

Anther development is the programmed process of stamen maturation during which functional microspores or pollen grains are produced for reproduction; this process is a key stage of the plant life cycle and is regulated by a series of genes [29]. The aberrant expression of these genes can lead to male sterility [30]. Plant CMS is due to interactions between mitochondrial and nuclear genes contributing to anther development, involving multiple genes in a complex regulatory network [31,32,33]. Therefore, understanding the gene expression patterns during anther development can help to elucidate the molecular regulatory mechanisms of pollen abortion in sterile plant lines.

In this study, the cytoplasmic male sterile line 1258A and its maintainer line 1258B of *B.napus* were compared with respect to the cytological characteristics of anthers during pollen development, sucrose, soluble sugar, proline (Pro), MDA, ATP, SOD, and POD contents, and other physiological and biochemical characteristics to understand the basis of 1258A infertility. Buds of sterile lines and maintainer lines were collected before and after the onset of pollen abortion for a transcriptome analysis to identify the genes and pathways involved in this process. The results of this study improve our understanding of the physiological characteristics and molecular mechanisms underlying CMS in *B.napus* and provide useful information for the utilization of hybrid vigor in CMS lines.

## 2. Results

### 2.1. Morphological and Cytological Characterization of 1258A and 1258B

There were no significant differences in stamen size and shape between the sterile line 1258A and the maintainer line 1258B. The anthers of the sterile line were bluntly triangular, fleshy, and pollen-free, while the pollen of the maintainer line was intact (Figure 1). To accurately describe the period of sterility onset, we performed cytological observations of fertile and sterile anthers at different developmental periods. As shown in Figure 2, the fertile and sterile anthers at the pollen mother cell period possessed complete pollen sacs and normal pollen mother cell differentiation (Figure 2A,F). At the tetrad stage, tetrads were observed in both fertile and sterile anthers; however, at this time, larger vacuoles were observed in the tapetum of sterile anthers, while the tapetum of fertile anthers showed darker staining with a denser cytoplasm (Figure 2B,G). During the uninucleate pollen grain period, the fertile anther tomentum began to degrade for microspore development (Figure 2C). At this time, the sterile anther tapetum was completely vacuolated and the microspores exhibited larger vacuoles (Figure 2H). During the development of microspores into mature pollen grains, the fertile anther tapetum degraded to release large amounts of nutrients (Figure 2D). The cells of the sterile anther tapetum were highly vesicularized and expanded, squeezing the microspores toward the center of the pollen sac (Figure 2I). Appendix A shows the process of sterile anther tapetum disintegration, in which the cells of the tapetum were vacuolated and squeezed the microspores inward without nutrient release. The fertile anther tapetum was completely disintegrated after the pollen was fully matured, residual traces of the tapetum were visible, and the pollen sac was close to dehiscence (Figure 2E). However, no pollen was produced in the pollen sacs of sterile anthers, residues of failed microspore development were visible, and the epidermis and tapetum of pollen sacs were tightly connected, generally without dehiscence (Figure 2J). Thus, 1258A pollen development began to show abnormalities in the cells of the tapetum during the tetrad stage, with vesiculation of the tapetum cells during microspore development, followed by cell expansion and disintegration, resulting in the inability of microspores to develop into mature pollen.

### 2.2. Physiological and Biochemical Characteristics of Sterile and Maintenance Lines

We selected 1258A and 1258B buds at different stages of anther development, stripped the anthers, and determined the sucrose, soluble sugar, ATP, Pro, MDA, POD, and SOD contents (Figure 3). Buds were collected at 2 mm (tetrad stage, pre-abortion), 3 mm (uninucleate stage, abortion), and 4 mm (mature pollen grain stage, late abortion). The sucrose content of fertile anthers was higher than that of sterile anthers at all stages, with significant differences between fertile and sterile anthers at the tetrad stage and uninucleate stage (Figure 3A). The soluble sugar content was higher in fertile anthers than in sterile anthers, but there was no significant difference at the tetrad stage, and the soluble sugar contents of sterile anthers at the uninucleate stage and mature pollen grain stage were significantly lower than those of fertile anthers (Figure 3B). The ATP content of fertile anthers was higher than that of sterile anthers during development; however, the ATP content of sterile anthers was significantly lower than that of fertile anthers before the onset of anther abortion and in the period after the onset of abortion (Figure 3C). These results indicate that in plants with sterile anthers, energy deficiency begins before the tetrad stage and increases during microspore development. The Pro content in all sterile anthers was significantly lower than that in fertile anthers, and the difference increased during pollen development, with fertile anthers containing nine times more free proline than sterile anthers during the mature pollen grain period (Figure 3D). The MDA content in sterile anthers was significantly higher than that in fertile anthers during the development process from pollen mother cells to microspores, and the MDA content in fertile anthers increased rapidly and was significantly higher than that in sterile anthers at the mature stage of pollen development (Figure 3E). The POD content in fertile anthers was significantly higher than that in sterile anthers before abortion and during abortion onset until the mature stage of pollen development, while the POD content in sterile anthers increased continuously with pollen development (Figure 3F). The SOD content of sterile anthers was lower than that of fertile anthers during development, especially during pollen mother cell development into microspores (Figure 3G). The H_2_O_2_ content of sterile anthers varied little throughout pollen development and was significantly lower than that of fertile anthers, in which H_2_O_2_ increased as pollen matured (Figure 3H).

### 2.3. Transcriptome Sequencing and Differentially Gene Expression Analyses

According to cytological observations, 1258A pollen abortion occurs due to the inability of microspores to develop into mature pollen grains. Therefore, we collected small buds (SB, 2 mm) from the tetrad period, before the formation of mononuclear pollen grains, as well as medium buds (MB, 4 mm) during mature pollen grain development for transcriptome sequencing. Three biological replicates of samples from each period were extracted and cDNA libraries were constructed for sequencing after RNA extraction. After data filtering, 30.1–47.82 million clean reads were obtained. Data were analyzed using Hisat2. The proportion of clean reads that matched the reference genome sequence was 90.97–92.63%, and the proportion of reads that matched to the unique position of the reference genome sequence was 81.44–82.36% (Table 1).

Differentially expressed genes (DEGs) were identified using DESeq with the following screening criteria: |log2FC| ≥ 1 and *p* ≤ 0.05. A summary of the DEGs in SB and MB buds of sterile and maintainer lines are shown in Figure 4A. In the comparison between 1258A_SB and 1258A_MB, 5549 genes were up-regulated and 4695 genes were down-regulated. For 1258B_SB vs. 1258B_MB, 5025 genes were up-regulated and 4458 genes were down-regulated. For 1258A_SB vs. 1258B_MB, 6385 genes were up-regulated and 6263 genes were down-regulated. For 1258A_SB vs. 1258B_MB, 6385 genes were up-regulated and 6263 genes were down-regulated. For 1258A_SB vs. 1258B_SB, 5551 genes were up-regulated and 5546 genes were down-regulated. For 1258A_MB vs. 1258B_MB, 6385 genes were up-regulated and 6263 genes were down-regulated. Next, these up- and down-regulated DEGs were further analyzed using Venny (http://bioinfogp.cnb.csic.es/tools/venny/index.html, accessed on 17 September 2021). Because the pollen of the maintainer lines developed normally, we assumed that the DEGs in these samples were not related to pollen defects; accordingly, the DEGs intersecting those for the maintainer line 1258B_SB vs. 1258B_MB were excluded. As shown in Figure 4B, among the up-regulated genes, 3173 DEGs were shared between the sterile and maintainer lines at the SB (1258A_SB vs. 1258B_SB) and MB (1258A_MB vs. 1258B_MB) stages. The shared DEGs were further cross-tabulated with the DEGs of the sterile lines, revealing 80 up-regulated genes. Similarly, among the DEGs that were down-regulated, 90 were down-regulated in 1258A_SB vs. 1258B_SB, 1258A_MB vs. 1258B_MB, and 1258A_SB vs. 1258A_MB (Figure 4C). The levels of these shared DEGs differed significantly from those in the maintainer lines during pollen abortion in sterile lines and may be related to pollen abortion.

### 2.4. GO Functional and KEGG Pathway Enrichment Analyses of Differentially Expressed Genes

A GO analysis of DEGs revealed enrichment for various GO terms, including binding, catalytic activity, cellular process, metabolic process, and membrane in the SB stage (1258A_SB vs. 1258B_SB) of the sterile and maintainer lines (Figure 5A). Among the DEGs in the MB stage (1258A_MB vs. 1258B_MB) of the sterile and maintainer lines, there was enrichment for binding, catalytic activity, cellular process, metabolic process, and cells (Figure 5B). The DEGs in the SB and MB stages of the sterile lines were mainly enriched for the terms binding, catalytic activity, membrane, cellular process, and metabolic process (Figure 5C).

We performed a GO functional enrichment analysis of DEGs and the top five GO terms significantly enriched in the sterile line 1258A_SB vs. 1258A_MB were lipid transport (GO:0006869), lipid localization (GO:0010876), inositol-3-phosphate synthase activity (GO:0004512), inositol metabolic process (GO:0006020), and inositol biosynthetic process (GO:0006021) (Figure 6A).The top five GO terms that were significantly enriched in 1258B_SB vs. 1258B_MB before and after anther abortion in the maintainer line were cellular response to auxin stimulus (GO:0071365), auxin-activated signaling pathway (GO:0009734), transferase activity transferring acyl groups (GO:0016746), cellular response to hormone stimulus (GO:0032870), and hormone-mediated signaling pathway (GO:0009755) (Figure 6B). These results indicated that the GO subclassifications before and after anther abortion in sterile and maintainer lines differed substantially. Similarly, the GO terms at the SB stage (1258A_SB vs. 1258B_SB) for sterile and maintainer lines were molecular function (GO:0003674), FMN binding (GO:0010181), acid phosphatase activity (GO:0003993), transferase activity, transferring pentosyl groups (GO:0016763), and cell proliferation (GO:0008283) (Figure 6C). The top five GO terms that were significantly enriched in the MB phase after septogenesis (1258A_MB vs. 1258B_MB) were lipid localization (GO:0010876), lipid transport (GO:0006869), lipid metabolic process (GO:0006629), cellular lipid metabolic process (GO:0044255), and lipid biosynthetic process (GO: 0008610) (Figure 6D). The DEGs at the MB stage after the onset of abortion in sterile and maintainer lines were the same as those in sterile lines before and after abortion in terms of lipid localization and lipid transport. In addition, GO terms related to floral development, reproductive system development, reproductive structure development, and reproductive development were significantly enriched in 1258A_SB vs. 1258B_SB in the “Biological Process” category (Appendix A).

To obtain insight into the interactions by which the DEGs exert their biological functions, we performed a KEGG pathway enrichment analysis of DEGs in 1258A SB, 1258B SB, 1258A MB, and 1258B MB using KOBAS and the KEGG PATHWAY database. The KEGG term with the most DEGs in 1258A SB, 1258B SB, 1258A MB, and 1258B MB buds was metabolic pathways; however, different pathways were significantly enriched in sterile and maintainer lines and at different stages. The most significantly enriched pathway in the sterile line 1258A_SB vs. 1258A_MB was phenylpropanoid biosynthesis, followed by the cutin, suberine and wax biosynthesis, cyanoamino acid metabolism, starch and sucrose metabolism, and glycerolipid metabolism pathways (Figure 7A). The most significantly enriched pathways in the maintainer line 1258B_SB vs. 1258B_MB comparison were the biosynthesis of cutin, suberine and wax biosynthesis, plant hormone signal transduction, and biosynthesis of secondary metabolites (Figure 7B). We found that cutin, suberine and wax biosynthesis were common in both sterile and maintainer lines, and these pathway differences may be present during normal pollen development. Accordingly, we focused on DEGs in the phenylpropanoid biosynthesis, cyanoaminoacid metabolism, starch and sucrose metabolism, and glycerolipid metabolism pathways. At the SB stage of the anther before abortion, the most significantly enriched pathway in the sterile lines 1258A_SB vs. 1258B_SB was the biosynthesis of secondary metabolites, followed by metabolic pathways, plant hormone signal transduction, glutathione metabolism, and inositol phosphate metabolism (Figure 7C). In the MB stage, the most significantly enriched pathway for both sterile and maintainer lines 1258A_MB vs. 1258B_MB was still the biosynthesis of secondary metabolites, followed by metabolic pathways, glutathione metabolism, etc. (Figure 7D). The biosynthesis of secondary metabolites and glutathione metabolic pathways were detected before and after pollen abortion in both sterile and maintainer lines; we hypothesize that DEGs in these two metabolic pathways have important effects on pollen abortion.

### 2.5. Hierarchical Clustering Analysis of Differentially Expressed Genes Associated with Pre- and Post-Pollen Abortion Stages in Sterile and Maintainer Lines

Both sterile and maintainer lines showed significant enrichment for biosynthesis of secondary metabolites and glutathione metabolic pathways at the SB and MB stages of the anther. Accordingly, we visualized the DEGs using pheatmap in R.

A hierarchical clustering analysis of DEGs involved in the biosynthesis of secondary metabolites (Figure 8A) showed that there were more down-regulated in the sterile lines than in the maintainer lines, both at the SB stage before abortion occurred and at the MB stage after abortion. We note that seven of the down-regulated expressed genes were β-glucosidases (*LOC106440633*, *LOC106422467*, *LOC106415266*, *LOC106445114*, *LOC106446946*, *LOC106419470*, and *LOC106429220*) and nine were associated with cytochrome P450 (*LOC106371512*, *LOC106360381*, *LOC106440154*, *LOC106440059*, *LOC106348517*, *LOC106445489*, *LOC106431114*, *LOC106410467*, and *LOC106437049*). However, some genes were also up-regulated in sterile lines, including six genes at the MB stage encoding *B.napus* UDP glycosyltransferases (*LOC106437891*, *LOC106451488*, *LOC106451089*, *LOC106375828*, *LOC106451088*, and *LOC106451487*). We also found that two peroxidase genes (*LOC106369846* and *LOC106376069*) in sterile lines were up-regulated after pollen abortion (Figure 8A, Class V). The results of a hierarchical cluster analysis of DEGs involved in glutathione metabolism are shown in Figure 8B; 19 genes were down-regulated in the sterile lines at both the SB and MB stages (Figure 8B, Classes II and III), which was significantly more than the number in the maintainer lines. These down-regulated genes were present from pollen mother cell development to mature pollen grain formation.

### 2.6. Hierarchical Clustering Analysis of Differentially Expressed Genes in Pre- and Post-Defect Stages in Sterile Lines

A hierarchical clustering analysis of DEGs involved in phenylpropanoid biosynthesis showed that 10 genes in the sterile line Class I were up-regulated in the SB stage during the pre-abortion stage and several genes in Class III were expressed at abnormally high levels in the MB stage during the late abortion stage (Figure 9A). Three genes in Class IV (*LOC106388515*, *LOC106405584*, and *LOC106422467*) were expressed at low levels in the MB stage after abortion, while in Class V, *LOC106446946*, *LOC106437132*, *LOC106437124*, and *LOC106419020* all showed low expression levels in the pre- and post-abortion stage (Figure 9A). These were mostly genes related to *B.napus* β-glucosidase and peroxidase, associated with phytohormone activation and antioxidant synthesis. A hierarchical clustering analysis of DEGs involved in cyanoaminoacid metabolism showed that in Class I, *LOC106421218* and *LOC106413118* were up-regulated at the SB stage, while *LOC106422956* and *LOC106428191* showed lower expression levels (Figure 9B). Similar to *LOC106420486* in Class IV, several genes in Class II also showed abnormally high expression levels at the MB stage (Figure 9B). In contrast, *LOC106437124*, *LOC106446946*, and *LOC106419211* in Class III all showed low expression levels in the pre- and post-abortion stages (Figure 9B). Similar to the abnormally expressed genes in the phenylpropanoid biosynthetic pathway, these genes were related to β-glucosidase in *B.napus*.

The results of hierarchical clustering analysis of DEGs involved in starch and sucrose metabolism showed that seven genes in Class III had abnormally high expression levels in the pre-anther abortion period, namely *LOC106421218*, *LOC106399638*, *LOC106346734*, *LOC106440633*, *LOC106439029*, *LOC106413118*, and *LOC106356810* (Figure 9C). Seven genes in Class V had abnormally low expression levels, namely *LOC106422956*, *LOC106454372*, *LOC106350934*, *LOC106428191*, *LOC106382995*, *LOC106419470*, and *LOC106361841* (Figure 9C). In the post-abortion stage, we found a large number of up-regulated genes related to starch and sucrose metabolism. For example, a large number of genes in the sterile line Class II had abnormally high expression levels, and only three genes in Class I showed abnormally low expression (Figure 9C). Thus, genes related to starch and sucrose metabolism are mainly abnormally expressed in the late stage of pollen abortion. For DEGs involved in glycerolipid metabolism, the up-regulated expression of genes in Class II was mainly detected during the pre-abortion stage of anthers (Figure 9D). At the late abortion stage, three genes in Class III were down-regulated, while five genes in Class V showed higher gene expression levels (Figure 9D). The number of up-regulated genes involved in starch and sucrose metabolism and glycerolipid metabolism in buds before and after anther abortion in sterile lines was higher than the number of down-regulated genes.

## 3. Discusion

### 3.1. 1258A Pollen Abortion Is Associated with Non-Programmed Cell Death in the Tapetum

The innermost layer of cells in the wall of the pollen sac, the tapetum, has dense cytoplasm and rich organelles, and the cytoplasm contains abundant protein and rich nutrients, such as oil and carotenoids, which play important nutritional and regulatory roles in the development or formation of pollen grains. The morphological features of the degenerated tapetum during normal anther development are a result of programmed cell death [34,35,36,37]. Premature or delayed programmed cell death in the tapetum disrupts the supply of nutrients to microspores, leading to pollen abortion [29,38]. *SaNa-1A* sterility in the *B.napus* CMS system occurs during the tetrad period, and abortion is mainly explained by the impaired degradation of tapetum cells, which limits nutrition required by microspores [39]. During the tetrad period of *SaNa-1A* pollen development, large vesicles in the tapetum, dense cytoplasm, abundant lysosomes, disruption of organelles and nuclear membranes, abnormal tapetum cell development, and delayed tapetum degradation lead to pollen abortion [40]. 1258A pollen abortion is similar to *SaNa-1A* pollen abortion in that large vacuoles appear in the cells of the 1258A tapetum during the tetrad period, followed by the gradual vesiculation of cells and disruption of the organelles and nuclear membrane. Unlike *SaNa-1A* pollen abortion, 1258A tapetum cells expand, and the cell wall elongates radially to squeeze the microspores and encroach on the anther locules. Studies of CMS systems, such as radish, cabbage, *B.oleracea*, and *B.napus*, have shown that the abnormal development of chorionic tapetum cells is a common phenomenon in pollen abortion [39,40,41,42,43].

### 3.2. Cellular Energy Deficiency and Pollen Abortion during Pollen Development

Sugars are the basic source of energy for plant cells and include sucrose, glucose, fructose, and their derivatives. The energy required during anther development is derived from carbohydrates produced by photosynthesis, mainly sucrose [44]. Pollen abortion due to an insufficient sugar supply has been studied extensively, and sugar metabolism has important roles in plant pollen development and male fertility [45]. The ATP, sucrose, and soluble sugar contents during pollen development in 1258A were significantly lower than those in 1258B, and the ATP content during the tetrad stage of pollen development was insufficient. Proline is an amino acid with many roles; during pollen development, it provides nutrients and promotes pollen growth and development by cooperating with abundant substances, such as carbohydrates. The proline content in 1258A anthers was significantly lower than that in the maintainer lines from the tetrad period, and the gap between the proline content in the sterile lines and the maintainer lines increased during pollen development. This indicates that the pollen is already facing an energy deficiency before pollen defects, limiting normal pollen development. ATP is the most direct energy supply for intracellular biosynthesis and functions and plays an important role in plant growth and development [46]. In *B.napus* CMS, F1F0-ATPase activity was inhibited during pollen development in sterile lines and the ATP content was low, affecting normal pollen development [40]. The ATP content in the anthers of sterile lines was also lower than that of the maintainer lines in *Nsa* CMS, and insufficient energy led to pollen abortion [19].

### 3.3. ROS Levels and Antioxidant Defense Systems during Pollen Development

ROS is a general term for a class of chemically active molecules or ions with high oxidative activity, mainly including superoxide anions (O_2_^−^), hydrogen peroxide (H_2_O_2_), hydroxyl radicals (HO^−^), and nitric oxide [47]. Mitochondria are the main site of ROS production, and excessive ROS can cause damage to biomolecules, such as proteins, nucleic acids, and lipids, thus affecting the normal physiological and biochemical functions of plant cells [48]. MDA is usually considered an indicator of ROS-induced lipid peroxidation and oxidative stress in plants. The MDA content of sterile lines increases significantly throughout pollen development in CMS lines and is consistently higher than that of the maintainer lines with excessive ROS accumulation [29,38,40,48]. In this study, we also found that the MDA content during 1258A pollen development was significantly higher in the tetrad and mononuclear pollen grains than corresponding levels in the maintainer lines, indicating that the microspores were affected by oxidative stress during microspore formation. The MDA content of the maintainer lines was higher than that of the sterile lines during the pollen maturation period, and the MDA content was reduced because the microspores of the sterile lines were deficient, and only residual material remained during this period. However, in the maintainer line, pollen development and the normal degradation of tapetum cells are accompanied by anther cell senescence, resulting in a large amount of MDA, which is a normal physiological phenomenon in anther development. In *B.napus SaNa-1A* CMS and *Nsa* CMS systems, high ROS accumulation during anther development is an important cause of pollen abortion [19,40].

Organisms have various systems for scavenging ROS, including SOD, catalase, glutathione, peroxidase, and ascorbic acid. This ensures that ROS levels in plants are not harmful to the organism. The main role of these antioxidant enzymes is to scavenge, in a timely manner, oxygen radicals and other products that are not conducive to cellular metabolism and are continuously produced, there by maintaining the balance of intracellular oxygen radicals and other systems [48]. Our results indicate that POD and SOD activity levels in 1258A are significantly lower than those of 1258B during microspore abortion, which may reduce the clearance of excessive ROS and impede microspore development. 1258A pollen does accumulate excessive ROS during development, as evidenced by a significantly higher MDA content than that of 1258B and significantly lower antioxidant enzyme activity than that of 1258B. Notably, the H_2_O_2_ level of 1258A was significantly lower than that of 1258B throughout pollen development, while the 1258B H_2_O_2_ content gradually increased with pollen maturation and anther senescence. We speculate that ROS accumulated in 1258A anthers may be other types, such as superoxide and hydroxyl radicals. ROS are produced under energy-intensive conditions that require vigorous metabolism; however, 1258A shows an energy deficiency from the beginning of pollen development, which may have an effect on hydrogen peroxide production. An imbalance between ROS accumulation and antioxidant defense systems are clearly an important cause of CMS pollen abortion. During wheat CMS pollen development, excessive ROS accumulation, reduced antioxidant enzyme activity, and disruption of the antioxidant system lead to the delayed initiation of programmed cell death in the tapetum, which eventually leads to pollen abortion and male sterility [38]. Studies of a kenaf CMS have also revealed that ROS accumulation is associated with cellular abnormalities in the tapetum [29]. Recent in-depth studies of the mechanism underlying CMS abortion in *B.napus* have shown that ROS accumulation in anthers is closely related to pollen abortion. However, various factors can cause excessive ROS accumulation in *B.napus* anthers. Defects in the antioxidant enzyme system during *SaNa-1A* CMS anther development exacerbate membrane lipid oxidation and excessive ROS accumulation, leading to MDA accumulation in the anther to poison the tapetum cells [40]. *Nsa* CMS pollen abortion is due to the production of toxic proteins by the mitochondrial gene *orf346*, leading to ROS over-accumulation in anthers [19].

### 3.4. 1258A Pollen Abortion Is Closely Associated with the Expression of Genes Related to Energy and Lipid Biosynthesis

A clustering analysis revealed genes in differential metabolic pathways that are closely related to energy biosynthesis pathways. Sucrose is the main product of photosynthesis and is then transported to lower tissues. After uptake by the cell, sucrose is incorporated into glucose (Glc), fructose, and uridine diphosphate Glc (UDPGlc) via invertases (INV) and sucrose synthase (SUS) and is further phosphorylated via Hexokinase (HXK1) to produce Glc-6-phosphate (Glc6P), resulting in Glc1P and trehalose-6-phosphate (T6P), which are involved in energy supply and signal transduction [49]. We found significant differences between sterile and maintainer lines in a large number of genes related to sugar metabolism and energy metabolism in the biosynthetic pathways of secondary metabolites. We noticed that several β-glucosidase genes (*LOC106440633*, *LOC106422467*, *LOC106415266*, *LOC106445114*, *LOC106446946*, *LOC106419470*, and *LOC106429220*) were down-regulated in the sterile lines, which may directly affect glucose generation and energy availability. In the case of energy deficiencies, genes encoding UDP glycosyltransferases (*LOC106437891*, *LOC106451488*, *LOC106451089*, *LOC106375828*, *LOC106451088*, and *LOC106451487*) are upregulated after pollen defeat in 1258A. Uridine diphosphate glucose (UDPG) is one of the most important glycosyl donors for the biosynthesis of glycosyl-containing compounds, such as oligosaccharides, polysaccharides, and glycoproteins [50]. There is evidence for a strong link between glucose signaling and ROS, with an association between pollen abortion and an inadequate energy supply and excessive ROS accumulation [29,40,51,52]. Glutathione is an important metabolite regulator involved in the tricarboxylic acid (TCA) cycle and gluconeogenesis, promoting carbohydrate and lipid metabolism [53]. In contrast, 19 DEGs involved in glutathione metabolism in sterile lines were down-regulated, which may reduce carbohydrate and lipid metabolism and affect the energy supply.

The phenylpropanoid biosynthesis pathway has an important role in pollen development. Genes in this pathway are associated with the outer wall of plant cells and are involved in the formation of sporopollenin proteins [54,55]. A KEGG pathway analysis showed that DEGs in these sterile lines were preferentially clustered in the “phenylpropanoid biosynthesis” pathway. Phenylalanine aminolytic enzyme 1 is the first key enzyme in the phenylpropane synthesis pathway; however, the gene encoding the phenylalanine aminolytic enzyme (*LOC106388515*) was significantly down-regulated. We also found that multiple genes encoding β-glucosidase and indole glucosinolate *O*-methyltransferase (*LOC106399204* and *LOC106356418*) were up-regulated and changes in the expression levels of these genes may affect the synthesis of phenanthrene and thus lignin and sporopollenin accumulation. Interestingly, we found that DEGs related to cyanoaminoacid metabolism showed similar expression patterns to those of the aberrantly expressed genes in the phenylpropanoid biosynthesis pathway, and these were genes related to *B.napus* β-glucosidase. In general, the functions and activity of β-glucosidases depend on the substrate specificity, tissue localization, cellular localization, and substrate interactions. In plants, β-glucosidases are involved in various physiological processes in plants, such as cell wall lignification, hormone metabolism in plants, and plant defense responses to adverse conditions [56,57,58,59,60].

During reproductive development, sucrose must be transferred from photosynthetically active tissues to reproductive organs via the siliques and transported to the developing pollen, where it is bound and stored as starch granules [61]. We detected the up-regulation of DEGs related to starch and sucrose metabolism in sterile lines after pollen abortion, suggesting that the energy deficit in microspore development induces an increased expression of sugar metabolism genes. Lipids are also important energy reserve substances. Similar to the starch and sucrose metabolic pathways, the expression levels of DEGs in the lipid metabolic pathway also change after pollen degradation, with a greater number of up-regulated expressed genes than down-regulated genes. The expression of genes related to energy and lipid biosynthesis was abnormal in sterile lines compared to levels in the maintainer lines, which may ultimately affect pollen sterility.

### 3.5. Cytochrome P450 Has an Important Role in Pollen Development

Plant cytochrome P450 monooxygenases (CYPs) are a large superfamily of hemesulfoxidases involved in many primary and secondary metabolic reactions. Members of the CYP86, CYP703, and CYP704 families are involved in the biosynthesis of anther cuticles and the outer wall of pollen [62]. A genome-wide analysis of cytochrome P450 genes revealed that *BoCYP* genes in kale are important for reproductive development [63,64]. A cytochrome P450 gene, *CYP704B* (also named *BoCYP704B1*) detected mainly in flower buds is expressed at significantly lower levels in sterile buds rather than in fertile buds [63]. Further functional studies have shown that *BoCYP704B1* functions in the *B. oleracea* tapetum and plays an important role in pollen development; mutations in this gene can lead to pollen abortion [64]. CYPs are essential for various biological processes. We found that nine cytochrome P450 genes were significantly down-regulated during 1258A anther development. Thus, 1258A pollen abortion may be closely associated with low cytochrome P450 gene expression, and these may be candidate genes for further studies of pollen abortion in *B.napus*.

## 4. Materials and Methods

### 4.1. Plant Materials

The *B.napus* sterile line 1258A and its homozygous heterozygous maintainer line 1258B were provided by Chunyun Guan. 1258A was a stable sterile line selected from the backcross population after generations of backcross selection using XinJiang *S.arvensis* as the parent and Xiangyou15 (*Brassica napus* L.) as the father. The above material was planted at the experimental base of Cultivation Park, Hunan Agricultural University, Changsha, Hunan, China.

### 4.2. Morphological and Cytological Observation

At the flowering stage, flowers of 1258A and 1258B blooming at the same time were collected. The petals were peeled off and placed under a stereomicroscope to observe morphological differences between the stamens of the sterile and maintainer lines. Semi-thin sections were made from 1258A and 1258B anthers of different sizes for observation. The procedure was as follows:

1. Harvest tissue block and fixation: Targeted fresh tissues should be selected to minimize mechanical damage such as pulling, contusion, and extrusion. Use a sharp blade to cut and harvest fresh tissue blocks quickly within 1–3 min. The size of tissue block should be no more than 1 mm^3^. Before sampling, petri dishes with fixative for TEM should be prepared in advance, small tissue blocks could be removed from animal body and immediately put into petri dishes, and then cut into small size of 1 mm^3^ in the fixative. The 1 mm^3^ tissue blocks were transferred into an EP tube with fresh TEM fixative for further fixation, meanwhile, keep vacuum extraction until the samples sink to the bottom. The samples were fixed for 2 h at room temperature and then fixed at 4 °C for preservation and transportation. Then, wash the tissues using 0.1 M PB (pH 7.4) for 3 times, 15 min each.

2. Post-fix: Tissues avoid light post fixed with 1% OsO_4_ in 0.1 M PB (pH 7.4) for 7 h at room temperature. After removing OsO_4_, the tissues are rinsed in 0.1 M PB (pH 7.4) for 3 times, 15 min each.

3. Dehydrate at room temperature, as follows: 30% ethanol for 1 h; 50% ethanol for 1 h; 70% ethanol for 1 h; 80% ethanol for 1 h; 95% ethanol for 1 h; 100% ethanol for 1 h; 100% ethanol for 1 h; ethanol:acetone = 3:1 for 0.5 h; ethanol:acetone = 1:1 for 0.5 h; ethanol:acetone = 1:3 for 0.5 h; pure acetone for 1 h.

4. Resin penetration and embedding as followed: acetone:EMBed 812 = 3:1 for 2–4 h at 37 °C; acetone:EMBed 812 = 1:1 overnight at 37 °C; acetone:EMBed 812 = 1:3 for 2–4 h at 37 °C; pure EMBed 812 for 5–8 h at 37 °C; pour the pure EMBed 812 into the embedding models and insert the tissues into the pure EMBed 812, and then keep in a 37 °C oven overnight.

5. Polymerization: The embedding models with resin and samples were moved into 65 °C oven to polymerize for more than 48 h. Then the resin blocks were taken out from the embedding models for standby application at room temperature.

6. Semi-thin section: The resin blocks were cut to 1 μm thin on the semi-thin slicer, and the tissues were fished out onto the microscope slides.

7. Toluidine blue staining: Keep toluidine blue dye solution in an oven at 60 °C for 1 h, and then stain the slides in the dye solution for 2 min. Wash the slides with running water, differentiate with 95% alcohol, and control the color under a light microscope (Olympus CX51, Olympus Corporation, Tokyo, Japan). Keep the slides in the oven to dry and then cover the slides with neutral resin.

### 4.3. Determination of Physiological and Biochemical Indexes

The contents of sucrose, soluble sugar, ATP, Pro, MDA, POD, and SOD in anthers were measured in buds of 1258A and 1258B at three stages corresponding to key periods in pollen development: tetrad period, microspore period, and mature pollen grain period. The buds were peeled with sterile forceps, and 0.1 g of intact anthers was quickly clamped, rapidly frozen in liquid nitrogen, and stored at −80 °C. The assay kits for the determination of sucrose, soluble sugar, ATP, Pro, MDA, POD, and SOD contents were purchased from Sangon Biotech Co. (Shanghai, China).

### 4.4. RNA Isolation, cDNA Library Preparation and Sequencing

Total RNA was extracted with Trizol (Invitrogen Corporation, Carlsbad, CA, USA) and assessed with Agilent 2100 BioAnalyzer (Agilent Technologies, Santa Clara, CA, USA) and Qubit Fluorometer (Invitrogen). Total RNA samples that meet the following requirements were used in subsequent experiments: RNA integrity number (RIN) > 7.0 and a 28S:18S ratio > 1.8. RNA-seq libraries were generated and sequenced by CapitalBio Technology (Beijing, China). The triplicate samples of all assays were constructed in an independent library and do the following sequencing and analysis. The NEB Next Ultra RNA Library Prep Kit for Illumina (NEB, Beijing, China) was used to construct the libraries for sequencing. NEB Next Poly(A) mRNA Magnetic Isolation Module (NEB) kit was used to enrich the poly(A) tailed mRNA molecules from 1 μg total RNA. The mRNA was fragmented into ~200 base pair pieces. The first-strand cDNA was synthesized from the mRNA fragments reverse transcriptase and random hexamer primers, and then the second-strand cDNA was synthesized using DNA polymerase I and RNaseH. The end of the cDNA fragment was subjected to an end repair process that included the addition of a single “A” base, followed by ligation of the adapters. Products were purified and enriched by polymerase chain reaction (PCR) to amplify the library DNA. The final libraries were quantified using KAPA Library Quantification kit (KAPA Biosystems, Cape Town, South Africa) and an Agilent 2100 Bioanalyzer. After quantitative reverse transcription-polymerase chain reaction (RT-qPCR) validation, libraries were subjected to paired-end sequencing with pair end 150-base pair reading length on an Illumina NovaSeq sequencer (Illumina, San Diego, CA, USA).

### 4.5. RNA-Seq: Data Analysis

The genome of *Brassica napus* reference genome Bra_napus_v2.0 was used as reference. The sequencing quality were assessed with FastQC (version 0.11.5) and then low quality data were filtered using NGSQC (version 2.3.3) [65]. The clean reads were then aligned to the reference genome using HISAT2 (version 2.1.0) [66] with default parameters. The processed reads from each sample were aligned using HISAT2 against the reference genome. The gene expression analyses were performed with StringTie (version 1.3.3b) [67]. DESeq (version 1.28.0) [68] was used to analyze the DEGs between samples. Thousands of independent statistical hypothesis testings were conducted on DEGs separately. Then, a *p*-value was obtained, which was corrected by FDR method. Corrected *p*-value (*q*-value) was calculated by correcting using the BH method. The *p*-value or *q*-value were used to conduct significance analysis. Parameters for classifying significantly DEGs are ≥2-fold differences (|log2FC| ≥ 1, FC: the fold change of expressions) in the transcript abundance and *p* ≤ 0.05. The annotation of the DEGs were performed based on the information obtained from the database of ENSEMBL, NCBI, Uniprot, GO, and KEGG.

### 4.6. GO and KEGG Pathway Enrichment Analysis

GOseq was used for the GO functional analysis of differentially expressed genes, including the GO functional enrichment analysis of differentially expressed genes and GO functional clustering, using the Gene Ontology database (http://www.geneontology.org/, accessed on 16 October 2021). KEGG enrichment analysis of differentially expressed genes was performed using KOBAS and the KEGG (http://www.genome.ip/kegg/, accessed on 17 October 2021) database.

### 4.7. Quantitative Real-Time PCR (qRT-PCR) Validation

To verify the accuracy of DEG data obtained by RNAseq, the relative expression levels of 12 key DEGs were analyzed by qRT-PCR (S2). Fluorescent quantitative primers for DEGs and *B.napus Actin2.1* (internal control) were designed on NCBI (https://www.ncbi.nlm.nih.gov/, accessed on 22 November 2021) and are listed in Appendix A. Relative expression was calculated with reference to the method of Livak et al. [69].

## 5. Conclusions

In this study, the cytological characteristics, physiological basis, and molecular mechanism of pollen abortion in the *B.napus* CMS line 1258A were revealed. A comparative analysis of the sterile line 1258A and the maintainer line 1258B revealed that pollen abortion in 1258A CMS is associated with non-programmed cell death in the tapetum, resulting in the failure of microspores to develop into mature pollen grains. The energy supply in sterile lines was insufficient in the early stage of anther development, and the contents of sucrose, soluble sugar, ATP, and Pro were significantly lower than those of the maintainer lines, contributing to pollen abortion. At the same time, intense oxidative stress occurred during pollen development in the sterile lines, with large amounts of MDA production, reduced activity of SOD and POD antioxidant enzymes, and dysfunction of cellular antioxidant defense system, resulting in pollen abortion. DEGs related to the biosynthesis of secondary metabolites, glutathione metabolism, phenylpropanoidbiosynthesis, cyanoaminoacid metabolism, starch and sucrose metabolism, and glycerolipid metabolism pathways may be involved in pollen abortion. Additionally, DEGs related to energy biosynthesis were significantly enriched, and the abnormal expression of these genes may affect cellular oxidative system homeostasis. The results of this study will help to elucidate the molecular mechanism underlying pollen abortion in the *B.napus* CMS line 1258A and provide a theoretical basis for better heterosis utilization in the species.

## Figures and Tables

**Figure 1 ijms-23-02004-f001:**
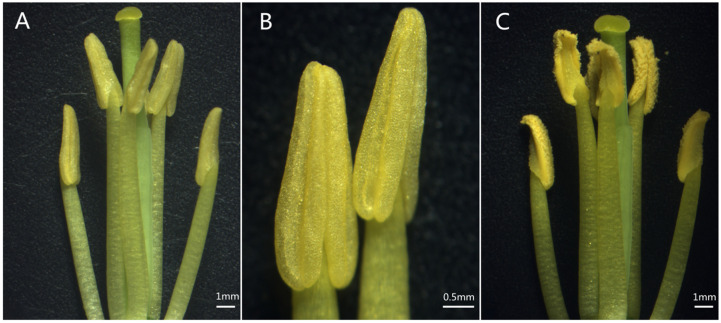
Phenotypic characterization of fertile and sterile flowers. (**A**) Partially dissected 1258A flower, Scale bar: 1 mm; (**B**) 1258A stamen, Scale bar: 0.5 mm; (**C**): partially dissected 1258B flower, Scale bar: 1 mm.

**Figure 2 ijms-23-02004-f002:**
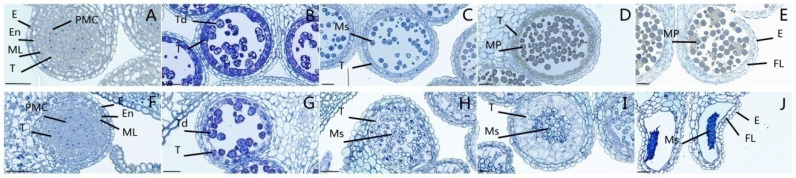
Cytological characteristics of 1258A (**F**–**J**) and 1258B (**A**–**E**) during pollen development. (**A**,**F**) PMC stage; (**B**,**G**) tetrad stage; (**C**,**H**) uninucleate stage; (**D**,**I**) mature pollen stage; (**E**,**J**) pollen release stage. E, epidermis; En, endothecium; FL, fibrous layer; ML, middle layer; MP, mature pollen; T, tapetum; Td, Tetrad; Ms, microspore; PMC, pollen mother cells; V, vascular region. Bar = 20 μm.

**Figure 3 ijms-23-02004-f003:**
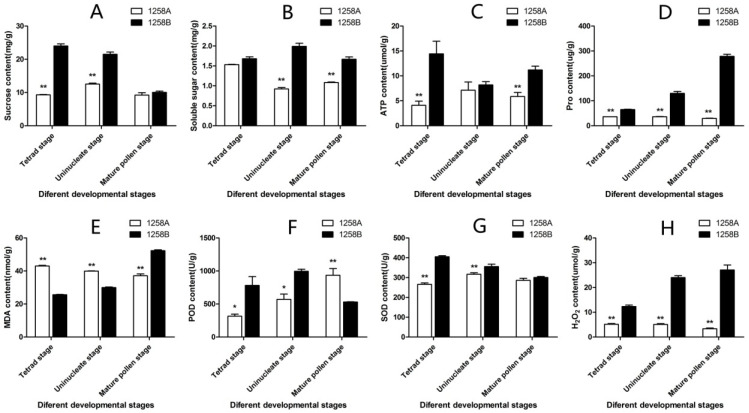
Physiological and biochemical indexes for 1258A and 1258B anthers at different periods of development. (**A**) Sucrose content; (**B**) soluble sugar content; (**C**) ATP content; (**D**) Pro content; (**E**) MDA content; (**F**) POD content; (**G**) SOD content; (**H**) H_2_O_2_ content; * *p* < 0.05 and ** *p* < 0.01.

**Figure 4 ijms-23-02004-f004:**
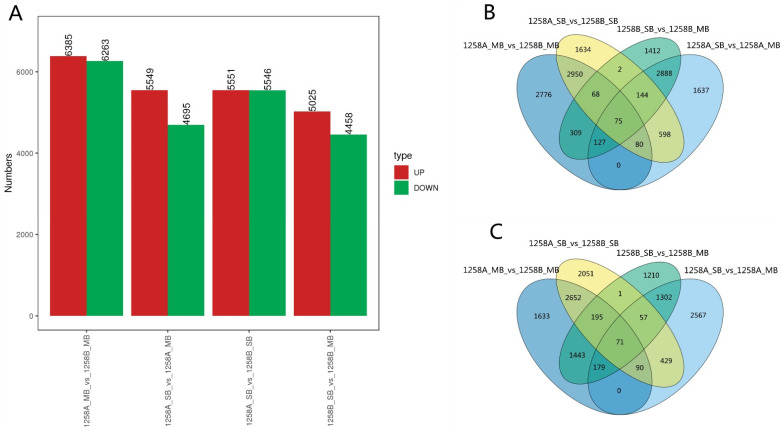
Overview of differentially expressed genes at the SB and MB stages of pollen development in 1258A and 1258B. (**A**) Number of differentially expressed genes; (**B**) up-regulated genes; (**C**) down-regulated genes.

**Figure 5 ijms-23-02004-f005:**
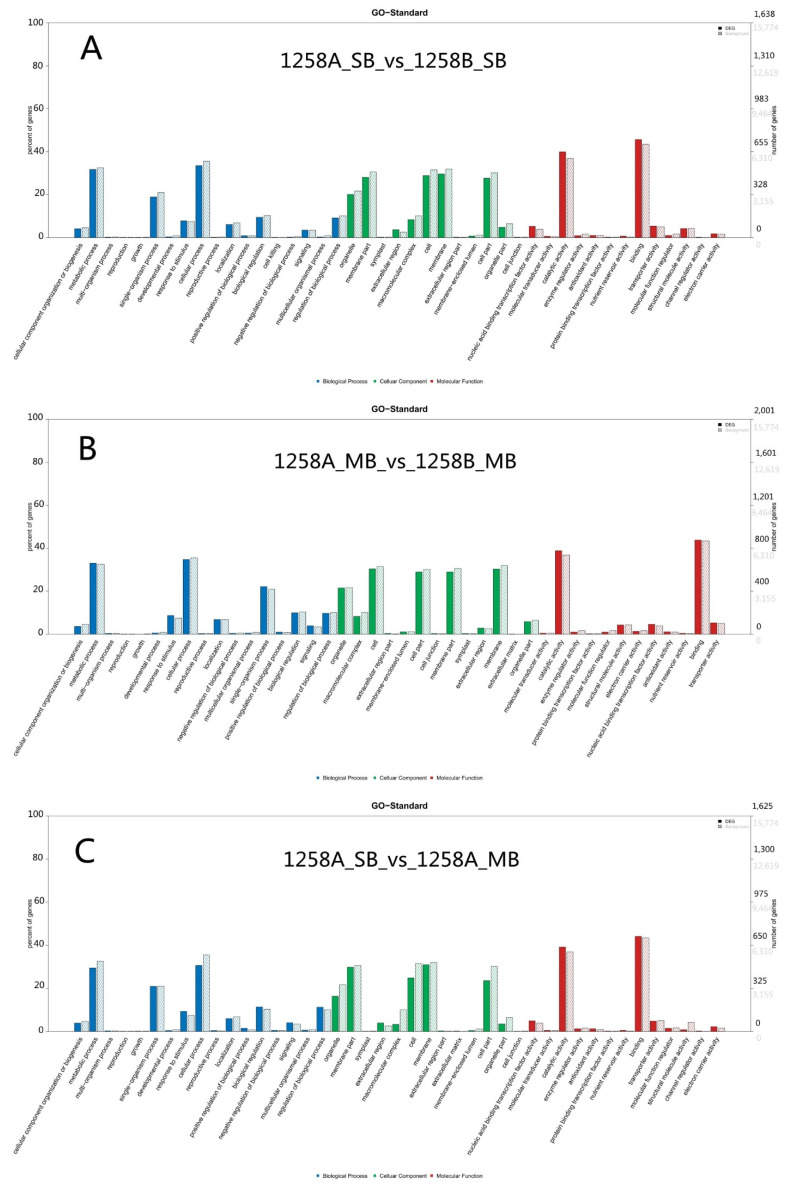
GO annotation analysis of differentially expressed genes in 1258A SB, 1258B SB, 1258A MB, and 1258B MB flower buds. (**A**) Number of differentially expressed genes for GO functions in 1258A_SB_vs_1258B_SB; (**B**) number of differentially expressed genes for GO functions in 1258A_MB_vs_1258B_MB; (**C**) number of differentially expressed genes for GO functions in 1258A_SB_vs_1258A_MB. Note: The horizontal axis shows the GO terms, the vertical axis shows the percentage of genes on the left and the number of genes on the right. This figure shows the distribution of genes for each secondary function of the GO database for the differential gene background and the all-gene background, reflecting the status of each secondary function in both backgrounds.

**Figure 6 ijms-23-02004-f006:**
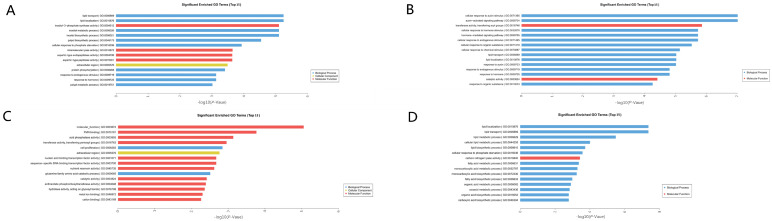
GO functional enrichment analysis of differentially expressed genes in 1258A SB, 1258B SB, 1258A MB, and 1258B MB buds. (**A**) 1258A SB vs. 1258A MB top 15 enriched entries; (**B**) 1258B SB vs. 1258B MB top 15 enriched entries; (**C**) 1258A SB vs. 1258B SB top 15 enriched entries; (**D**) 1258A MB vs. 1258B MB top 15 enriched entries. Note: In the bar graphs, larger values along the horizontal axis indicate greater enrichment for the term. The top 15 terms with the smallest corrected *p*-value/*p*-value in the enrichment results are shown.

**Figure 7 ijms-23-02004-f007:**
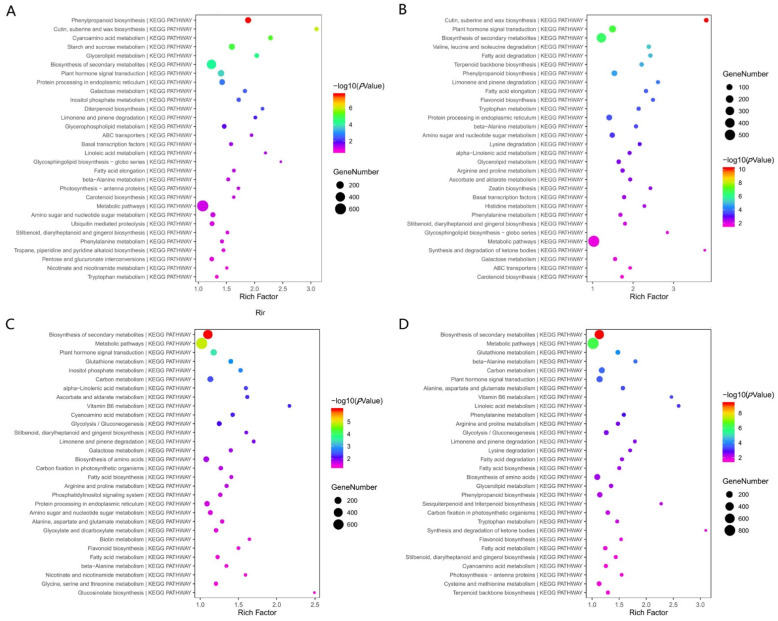
KEGG pathway enrichment analysis of differentially expressed genes in 1258A SB, 1258B SB, 1258A MB, and 1258B MB flower buds. (**A**) 1258A_SB vs. 1258A_MB differentially expressed gene KEGG pathway enrichment; (**B**) 1258B_SB vs. 1258B_MB differentially expressed gene KEGG pathway enrichment; (**C**) 1258A_SB vs. 1258B_SB differentially expressed gene KEGG pathway enrichment; (**D**) 1258A_MB vs. 1258B_MB differentially expressed gene KEGG pathway enrichment. Note: The horizontal axis indicates the number of annotated differential genes/number of annotated background genes in the pathway, and a larger value indicates a greater enrichment. The color represents the *p*-value of the significance test, where smaller values indicate greater enrichment. The size of the circle represents the number of differential genes.

**Figure 8 ijms-23-02004-f008:**
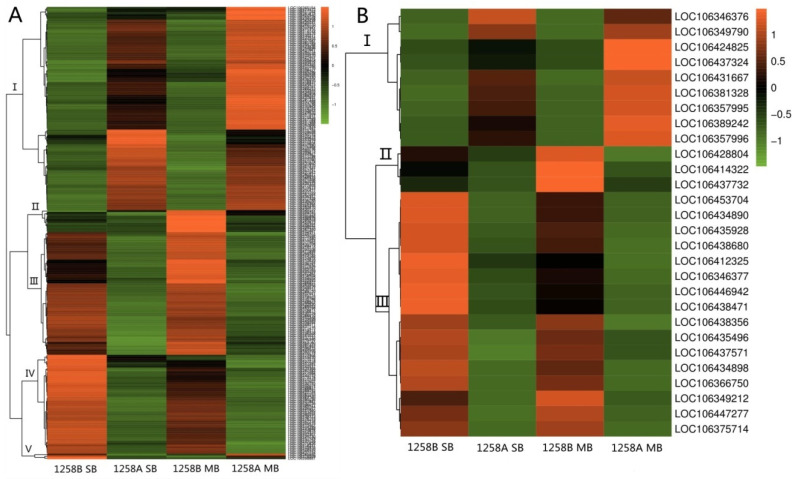
Hierarchical clustering analysis of differentially expressed genes in the SB and MB stages of the anther enrichment pathway in sterile and maintainer lines. (**A**) Hierarchical clustering analysis of differentially expressed genes involved in the biosynthesis of secondary metabolites; (**B**) hierarchical clustering analysis of differentially expressed genes involved in glutathione metabolism.

**Figure 9 ijms-23-02004-f009:**
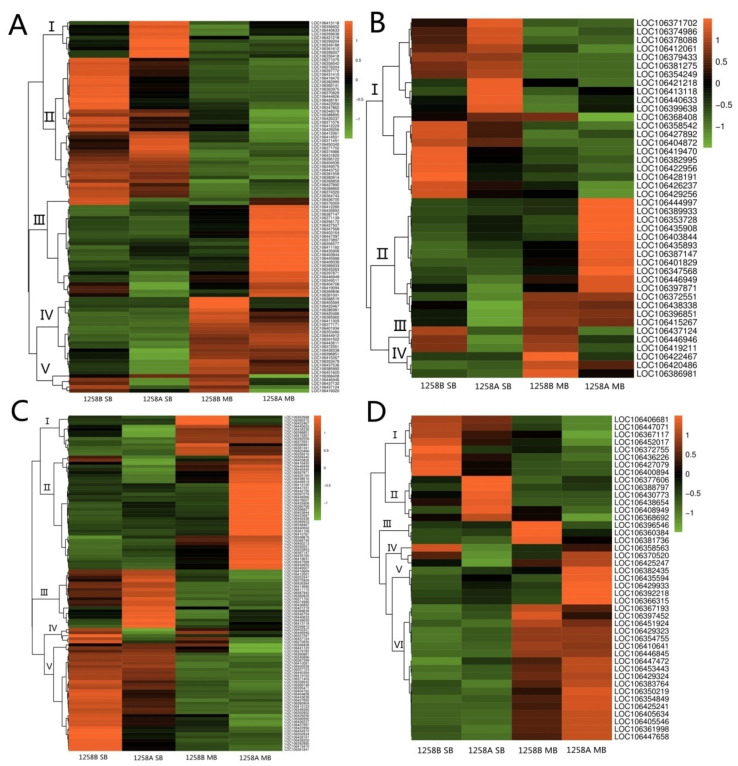
Hierarchical clustering analysis of differentially expressed genes in 1258A at the anther SB and MB stages. (**A**) Hierarchical clustering analysis of differentially expressed genes involved in phenylpropanoid biosynthesis; (**B**) hierarchical clustering analysis of differentially expressed genes involved in cyanoaminoacid metabolism; (**C**) hierarchical clustering analysis of differentially expressed genes involved in starch and sucrose metabolism; (**D**) hierarchical clustering analysis of differentially expressed genes involved in glycerolipid metabolism.

**Table 1 ijms-23-02004-t001:** Summary of RNA-seq data.

Samples	Replicates	Raw Read Count	Clean Reads	Clean Rate (%)	Mapped Reads	Uniquely Mapped Reads	Multiply Mapped Reads
1258A_SB	1258A_SB1	30,803,764	30,107,242	97.26	27,389,981 (90.97%)	22,348,065 (81.59%)	5,041,916 (18.41%)
	1258A_SB2	46,866,900	45,821,274	97.3	41,915,526 (91.48%)	34,230,714 (81.67%)	7,684,812 (18.33%)
	1258A_SB3	43,480,702	42,561,492	97.43	38,898,424 (91.39%)	31,776,767 (81.69%)	7,121,657 (18.31%)
1258B_SB	1258B_SB1	47,799,244	46,717,546	97.34	43,014,713 (92.07%)	35,097,253 (81.59%)	7,917,460 (18.41%)
	1258B_SB2	46,655,566	45,567,026	97.25	41,951,909 (92.07%)	34,180,575 (81.48%)	7,771,334 (18.52%)
	1258B_SB3	36,884,696	36,172,110	97.69	33,461,220 (92.51%)	27,251,109 (81.44%)	6,210,111 (18.56%)
1258A_MB	1258A_MB1	48,872,338	47,823,512	97.41	43,878,595 (91.75%)	36,042,663 (82.14%)	7,835,932 (17.86%)
	1258A_MB2	47,453,454	46,448,980	97.46	42,534,489 (91.57%)	35,016,906 (82.33%)	7,517,583 (17.67%)
	1258A_MB3	48,333,726	47,497,528	97.91	43,558,465 (91.71%)	35,876,856 (82.36%)	7,681,609 (17.64%)
1258B_MB	1258B_MB1	40,097,894	39,154,206	97.22	36,083,330 (92.16%)	29,637,892 (82.14%)	6,445,438 (17.86%)
	1258B_MB2	48,660,014	47,635,164	97.52	44,125,656 (92.63%)	36,101,966 (81.82%)	8,023,690 (18.18%)
	1258B_MB3	36,587,760	35,863,484	97.63	33,141,893 (92.41%)	27,212,585 (82.11%)	5,929,308 (17.89%)

Mapped reads: reads that match to the reference genome (as a proportion of the total reads); Uniquely mapped reads: reads that match to a unique position in the reference genome sequence (proportion of mapped reads); Multiple mapped reads: reads that match to multiple positions in the reference genome sequence (as a proportion of the mapped reads).

## Data Availability

The data presented in this study are available on request from the corresponding author.

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
