# Peer review of "Comparative Cytological and Transcriptome Analyses of Anther Development in Nsa Cytoplasmic Male Sterile (1258A) and Maintainer Lines in Brassica napus Produced by Distant Hybridization"

_ijms, 2022, doi:10.3390/ijms23042004_

Round 1

Reviewer 1 Report

The research is relevant. The manuscript presents the results of comprehensive study aimed to clarifying cellular and molecular mechanisms of cytoplasmic male sterility manifestation in the line with Nsa CMS. The comparative study of a sterile CMS line and its fertile analogue was carried out at the cytological and molecular levels, with applying a complex approach involving morphological, cytological, biochemical, transcriptomic and bioinformatic methods. The results obtained have demonstrated dynamics in changes of a number key compounds important for pollen development. They also reveal significant alterations in the expression patterns of a number of genes encoding for cytochrome P450 genes that indicating their important role pollen abortion. The data are illustrated with one table, and 9 informative figures. A rich additional information is given in supplementary tables.

Comments.

  1. Text editing is required.
  2. It is necessary to edit the article title. The name of CMS type must be present in the title or at least be included in key words.
  3. The section Materials and Methods should be substantially improved to made the results more understandable:
  • To give the title and affiliation of the researcher who kindly presented plant material.
  • To describe in more details analysis of physiological and biochemical indices.
  • The details of transcriptomic analysis should be described in the section.
  • To indicate in the text the source of reference genome used in the analysis.

Author Response

Thank you very much for your comments! We have carefully revised it according to your suggestions. Edited the title of the article and added CMS type in the title. Improved content of materials and methods. See Chapter 4.2,4.4 and 4.5 for details. We add detailed methods for transcriptome analysis. The genome of Brassica napus reference genome Bra_napus_v2.0 was used as reference.

Reviewer 2 Report

The presented work is devoted to a very important topic, the study of the nature of cytoplasmic male sterility. A comparative analysis of the morphology, cytology, biochemistry and gene expression between the CMS line and maintainer line in Brassica napus was carried out. The manuscript is well structured and well written.

However, some revisions are required before the manuscript can be considered for publication.

There are no spaces between words in the manuscript in many places. Check it.

In a sentence, line 26-28, it looks like a predicate is missing.

Line: 391 Uniformity is needed. Please use Latin for all names.

Currently, there are various studies of the transcriptome of B.napus CMS lines, including Nsa CMS. What are the similarities or differences between them and your results? Please discuss this.

Author Response

Thank you very much for your comments! We have carefully revised the manuscript, and many words without spaces have been corrected. Compared to the Nsa CMS reported, our sterile line has many similarities and differences. In the discussion section, we discussed the results of 1258A study with Nsa CMS and other infertility studies. Transcriptome analysis found that 1258A Pollen Abortion is Closely Associated with the Expression of Genes Related to Energy and Lipid Biosynthesis.Therefore, we focused on the role of transcriptome analysis results in pollen abortion, and did not carry out comparative analysis with other CMS transcriptome data. Pollen abortion in Nsa CMS is closely related to mitochondrial genome. Next, we will further study the mitochondrial genome of 1258A.